# Grounding is All You Need? Dual Temporal Grounding for Video Dialog

## Abstract

In the realm of video dialog response generation, capturing both the essence of video content and the temporal nuances of conversation history is crucial. While some approaches rely on large-scale pretrained visual-language models, often neglecting temporal dynamics, others emphasize spatial-temporal relationships within videos but demand intricate object trajectory pre-extractions and overlook dialog temporal dynamics. This paper introduces the Dual Temporal Grounding-enhanced Video Dialog model (DTGVD), designed to bridge the gap between these two approaches. DTGVD uniquely integrates the strengths of both by emphasizing dual temporal relationships. It achieves this by predicting dialog turn-specific temporal regions, selectively filtering video content, and grounding responses in both video and dialog contexts. A key innovation of DTGVD is its advanced handling of chronological interplay within dialogs. By effectively capturing and leveraging dependencies between dialog turns, it enables a more nuanced understanding of conversational dynamics. To further align video and dialog temporal dynamics, we introduce a list-wise contrastive learning strategy. In this framework, accurately grounded turn-clip pairings are treated as positive samples, while less precise pairings serve as negative samples. This refined classification is then seamlessly integrated into our end-to-end response generation mechanism. Evaluations using AVSD@DSTC-7 and AVSD@DSTC-8 datasets underscore the superiority of our methodology. Our code[1] will be made public later.

## 1 Introduction

Video dialog aims to generate a free-form response to a follow-up question based on both the video content and the history of multi-turn question-answer pairs, as illustrated in Fig. 1. This task shares similarities with other vision-and-language tasks, such as video sentence grounding (Xiao et al., 2021; Liu et al., 2020; 2021b;a), video question answering (Li et al., 2022; Xiao et al., 2022), and video relation detection (Shang et al., 2021; Gao et al., 2022). However, unlike image-grounded dialog (Murahari et al., 2020; Chen et al., 2020), video dialog demands a more hierarchical understanding and reasoning process, encompassing complex elements like actions, events, and interactions, which inherently carry far richer information than static images. The primary challenge of video dialog lies in accurately comprehending the dynamic content of the video while simultaneously leveraging the evolving dialog history between the user and the dialog agent. Effectively addressing these challenges is crucial for generating coherent, contextually relevant, and sensible responses.

Recent advances in video dialog have leveraged large-scale pretrained models such as GPT (Radford et al., 2019), UniVL (Luo et al., 2020), and LLaMA (Touvron et al., 2023). These models, fine-tuned to accept video frames, dialog history, and questions, have shown significant promise in addressing video dialog challenges. Their strength lies in utilizing extensive pre-existing knowledge, which helps mitigate the limitations of relatively small video dialog datasets. However, despite their success in vision-and-language tasks, these models often struggle to capture temporal relationships in dialog history, resulting in inaccuracies by incorporating irrelevant video content (Le & Hoi, 2020; Li et al., 2021; Yamazaki et al., 2022), limiting their ability to exploit the temporal dynamics needed for coherent video dialog understanding. On the other hand, object-centric methods, such as those proposed by Geng et al. (2021), Kim et al. (2021), and Pham et al. (2022), attempt to capture temporal relationships by focusing on object trajectories extracted from video sequences using tools like Faster-RCNN and the DeepSort algorithm. These methods construct detailed spatial-temporal

---

[1] https://anonymous.4open.science/r/video_dialog-4EE6/

graphs and alignment strategies, yet they face challenges when multiple objects in a single video clip correspond to diverse question-answer pairs. Moreover, their computational demands can be substantial, making them less efficient, especially for complex video dialog tasks.

Given the dynamic nature of video dialog, it's crucial to enhance the granularity of temporal localization for each question-answer pair, which can significantly improve response generation. By capturing the interplay of related dialog turns, models can achieve richer context comprehension. However, many existing approaches (Pham et al., 2022; Shah et al., 2022), as shown in Fig. 1, have either relied solely on recent dialog turns or processed dialog history linearly, overlooking the distinct temporal relevance of each pair to the video content, thus missing opportunities for more accurate response generation.

Therefore, we introduce the Dual Temporal Grounding-enhanced Video Dialog (DTGVD) model. This innovative approach capitalizes on the dual temporal dynamics inherent in both video sequences and dialog histories. At its foundation, DTGVD employs the UniVL pretrained visual-language model (Luo et al., 2020) to discern the critical temporal segments of each dialog interaction. This allows for a focused response generation that is rooted in contextually relevant video segments while simultaneously leveraging pertinent dialog turns. The model's design exhibits a meticulous attention to the temporal intricacies of conversations. To further enhance this alignment, we incorporate a list-wise contrastive learning paradigm: accurately grounded turn-clip pairings are treated as positive benchmarks, guiding the system away from less accurate predictions. This strategy culminates in a comprehensive end-to-end training mechanism that prioritizes reference response fidelity. Overall, our main contributions are summarized as follows:

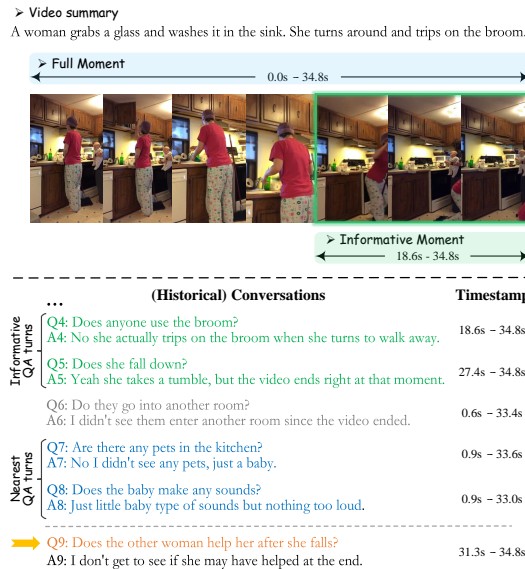

Figure 1: Given a video clip and dialog history (Q1&A1-Q8&A8), video dialog model generates the corresponding answer (A9) to the current question (Q9). Most previous methods merely exploit the nearest several turns of question-answer pairs (e.g., Q7&A7, Q8&A8) and Full Moment. In our method, we ground the temporal region of each QA pair in the video, and select Informative Moment and informative QA pairs for generating the responses (e.g., Q4&A4, Q5&A5).

- We propose a temporal grounding module to explicitly model the attention shift of each dialog turn over the video, and generate the temporal masks to filter out irrelevant video frames and irrelevant dialog history.

- Based on the predicted temporal region of each QA pair, we design a novel contrastive objective function to enhance the selection of related video clips.

- We achieve promising performance as compared with SOTA methods. Experiments on two popular benchmark datasets verified the effectiveness of our method. And experiments on various pretrained models verified the expandability of the method.

## 2 RELATED WORK

### 2.1 VIDEO DIALOG

Recently, with AVSD@DSTC-7 (Yoshino et al., 2019), AVSD@DSTC-8 (Kim et al., 2019) and AVSD-@DSTC-10 (Hori et al., 2022) challenges, Video-grounded Dialog (VGD) has received a lot of attention. As a crucial component of multi-modal reasoning tasks, VGD requires the model to comprehensively consider dialogue history, current query and video scenes to facilitate response generation. Early works Alamri et al. (2019); Chao et al. (2019); Hori et al. (2019a); Le et al.

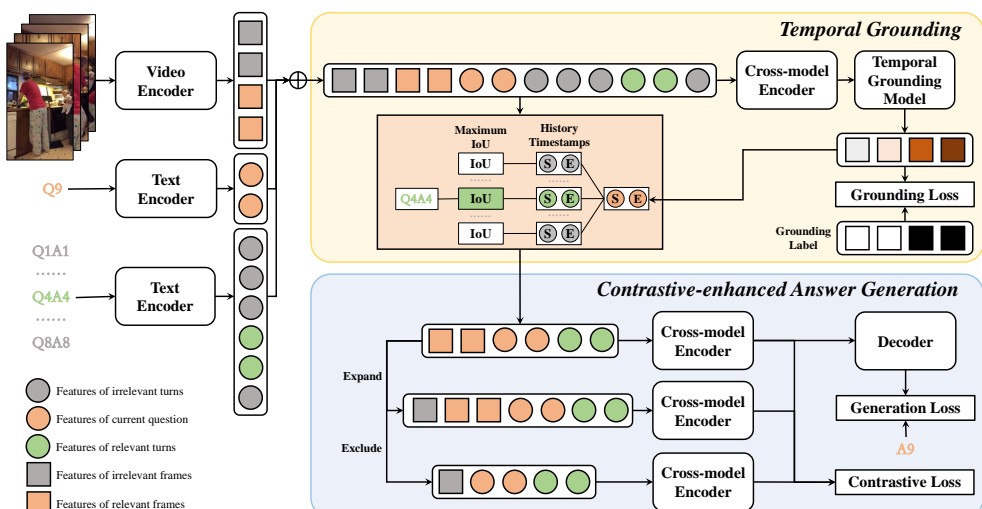

Figure 2: The pipeline of our proposed DTGVD is made up of four primary components including Basic Encoder, Temporal Grounding, Answer Generation and Contrastive Selection. The whole model is trained with a contrastive learning-based loss function and a text generation loss function. The symbol $\oplus$ means concatenating multi-model features along the time/sequence dimension.

(2019a); Nguyen et al. (2019); Sanabria et al. (2019) used recurrent neural network or multi attention to encode dialog and convolutional neural network to obtain video features, with simple concatenation for cross-modal fusion.

Subsequent researches are mainly divided into two groups: one group opts to utilizing visual-language pre-trained models. For example, Le & Hoi (2020) and Li et al. (2021) embedded video into text space and fined turn a GPT-2 (Radford et al., 2019) model to generate the answers. Yamazaki et al. (2022) employed a pre-trained TimeSformer (Bertasius et al., 2021) model to obtain better visual representation. Huang et al. (2022) applied an UniVL (Luo et al., 2020) model to enhance multi modal representation and fusion capabilities. Zhang et al. (2023) leverages the powerful text generation capability of Large Language Model (LLM) to convert videos into embeddings that LLaMA (Touvron et al., 2023) can recognize using Q-former. However, researches in this group have an insufficient utilization of features and generally ignore the temporal relationships between various modalities. For example, they input the entire video or several recent dialogue history turns. This results in abundance of noise that undermines the advantage of pre-trained models and hinders their effectiveness. The other group is object-centered that focuses on extracting spatial-temporal information relevant to objects from the video or text. For example, Geng et al. (2021) and Kim et al. (2021) obtained object features by Faster R-CNN (Ren et al., 2015) and constructed scene graphs to perform object-centric cross-modal interactions. Pham et al. (2022) parsed the dynamic space-time visual content into object trajectories and leveraged questions. However, these methods require complex pre-extraction of object trajectories and mainly focus on cross-modal fusion between vision and text, without fully utilizing the temporal relationships in conversation history. Besides, in the era of large models, they still need to train complex networks from scratch, which may soon be surpassed by a simple fine-tuned multi-modal pre-trained model. Our work addresses the issues of both groups, by extracting more effective key information from video and text based on temporal dependencies. Besides, our framework can work with a variety of pre-trained models, which demonstrates significant superiority in this task.

## 2.2 VIDEO TEMPORAL GROUNDING

Video temporal grounding (VTG) aims to pinpoint the start and end times of a target segment within an untrimmed video in relation to a given query. Early research Chen et al. (2018); Chen & Jiang (2019); Xu et al. (2019) mostly adopted a two-stage process. This involved first obtaining candidate segments, either through a sliding window or generated proposals, and then separately learning the representations of textual and visual content. The final step involved identifying specific time seg-

ments via classification and regression. Subsequent studies, however, shifted away from presenting candidates and instead directly determined the target start and end coordinates in an end-to-end fashion. Zhang et al. (2020a) and Yuan et al. (2019) utilized co-attention to fuse video and text features extracted from C3D and GloVe, and obtained the start and end timestamps through regression. Mun et al. (2020) obtained semantics-aware segment features based on the extracted phrase features via local-global video-text interactions. Zhang et al. (2020b) constructed a 2D temporal feature map to better retrieve video length candidates with different duration in an end-to-end manner.

It is evident that combining VTG with video reasoning tasks can lead to more refined video understanding. However, not many studies have delved into this area. Lei et al. (2019) developed a dataset that includes time segments corresponding to each question and answer and introduced a locate-then-answer VQA model. Meanwhile, Li et al. (2022) enhanced video answer accuracy by eliminating video clips that were irrelevant to the query in focus. A possible reason for the limited exploration of this combination is that most existing grounding models possess distinctive designs, making them challenging to seamlessly integrate into downstream task models. To address this issue, our DTGVD model incorporates a temporal grounding component. This component is designed to share partial weights and can seamlessly execute both the grounding and reasoning processes within a singular model.

## 3 METHOD

We introduce a Dual Temporal Grounding-enhanced Video Dialog model, named DTGVD, as shown in Fig. 2. We first provide the problem definition in Sec. 3.1, and introduce four main components of DTGVD, namely Basic Encoder, Temporal Grounding, Answer Generation and Contrastive Selection, from Sec. 3.2 to Sec. 3.5.

### 3.1 PROBLEM DEFINITION

Given an untrimmed video $V = \{v_t\}_{t=1}^T$ and the dialog history of $K-1$ turns of question-answer pairs $H_{K-1} = (Q_{1:K-1}, A_{1:K-1})$, where $T$ and $K$ are the number of frames and dialog turns, respectively, the goal of video dialog is to generate a free-form natural language answer $A_K$ of the question $Q_K$, which can be summarized as:

$$\hat{A}_K = \arg\max_A P\left(A \mid V, H_{K-1}, Q_K; \theta\right), \tag{1}$$

where $\theta$ is the parameter of video dialog model.

How to locate valuable information from the dialog history and video is a major challenge of this task, given the abundance of irrelevant and disruptive information present in the complete video and all previous turns. If we utilize $\mathcal{V}$ to indicate a subset of $V$ that contains the significant video frames, and $\mathcal{H}$ to indicate the set that includes effective history turns, the objective of the task can be simplified to:

$$\hat{A}_K = \arg\max_A P\left(A \mid \mathcal{V}, \mathcal{H}, Q_K; \theta\right). \tag{2}$$

Thus, the complicated task can be converted into two straightforward parts, which consist of *temporal grounding* to discover beneficial video clips with turns (i.e. $\mathcal{V}$ and $\mathcal{H}$), and *answer generation* to obtain accurate answer.

### 3.2 BASIC ENCODER

We employ basic text and video encoder to embed dialog history and the video respectively, following the structure of Univl (Luo et al., 2020).

**Text Encoder.** To process the input question and dialog history, we apply the BERT pre-processing procedure, resulting in a token sequence $\mathbf{t} = \{t_i \mid i \in [1, n]\}$, where $t_i$ refers to the $i$-th token and $n$ denotes the sequence length. Subsequently, we employ the BERT-based uncased model to generate the text representation $\mathbf{T} \in \mathbb{R}^{n \times d}$ by feeding the token sequence $\mathbf{t}$ into the model: $\mathbf{T} = \text{BERT}(\mathbf{t})$, where $d$ represents the hidden size of the textual representation.

**Video Encoder.** We extract features from a frame sequence $\mathbf{v} = \{v_j \mid j \in [1, T]\}$ for each video, where $v_j$ represents the $j$-th frame of the video and $T$ is the length of the frame sequence. A pretrained video feature extractor, S3D (Xie et al., 2018a), is used to generate the video feature $\mathbf{F}_v \in \mathbb{R}^{m \times d_v}$, where $m$ refers to the length of the time dimension and $d_v$ is the hidden size of the video features. We then apply a Transformer-based encoder to embed the contextual information of the video into $\mathbf{V} \in \mathbb{R}^{m \times d}$, formulated as $\mathbf{V} = \text{Transformer}(\mathbf{F}_v)$.

### 3.3 TEMPORAL GROUNDING

In this section, we aim to identify specific useful dialog turns and video clips via temporal dependencies.

**Cross-modal Encoder.** In order to facilitate full interaction between text and video, we utilize a Transformer-based cross-modal encoder to handle the concatenated feature. The cross-modal encoding feature $\mathbf{M}_{\mathbf{T}, \mathbf{V}} \in \mathbb{R}^{(n+m) \times d}$ can be expressed as follows:

$$\mathbf{M}_{\mathbf{T}, \mathbf{V}} = CrossEncoder(\mathbf{T} \oplus \mathbf{V}), \tag{3}$$

where $\oplus$ means concatenation operation. Note that the multi-modal features are concatenated along the time dimension of video and sequence dimension of text, which can be utilized easily to obtain the frame-level grounding results.

**Video Mask.** We explore the temporal relation between each QA turn and video, by predicting the start and end timestamp $(\tau_i^s, \tau_i^e)$ in the video corresponding to each question $Q_i$, where $i \in [1, K]$ and $(\tau_i^s, \tau_i^e) = f(V, H_{i-1}, Q_i; \theta)$.

Specifically, based on the cross-model representations $\mathbf{M}_{\mathbf{T}, \mathbf{V}}$, we use the part corresponding to $\mathbf{V}$ to predict the time mask:

$$V_i^{mask} = F(\mathbf{M}_{\mathbf{V}}), \tag{4}$$

where $V_i^{mask}$ represents the predicted temporal mask for question $Q_i$, $F$ represents the combination of a Conv1D layer and sigmoid activation function for mask prediction.

As for frame level, the temporal mask can also be treated as the binary classification result on whether each frame is relevant to current question. We apply binary cross-entropy (BCE) loss to measure the difference of predicted result and ground truth:

$$L_{\text{frame}} = \sum_{j=1}^{m} L_{\text{bce}}(P_i^j, Y_i^j), \tag{5}$$

where $Y_i^j$ is the label on whether frame $j$ is related to $Q_i$, and $P_i^j$ is the predicted result.

As for the segment level, we utilize cross-entropy (CE) loss to compare the predicted start and end timestamps with the label:

$$L_{\text{clip}} = \frac{1}{2} \big[ L_{\text{ce}}(p_i^s, t_i^s) + L_{\text{ce}}(p_i^e, t_i^e) \big], \tag{6}$$

where $t_i^s$ and $t_i^e$ are the labels of the start and end boundaries, respectively. $p_i^s$ and $p_i^e$ are the predicted values of the start and end timestamps. The final loss of temporal grounding can be represented as:

$$L_{\text{grounding}} = \lambda L_{\text{clip}} + L_{\text{frame}}, \tag{7}$$

where $\lambda$ is a hyperparameter to control the ratio of the two losses.

Then, we can generate the predicted timestamp $(\tau_i^s, \tau_i^e)$ of each question:

$$
\begin{aligned}
\tau_i^s &= \frac{1}{2} \big[ minIdx(V_i^{mask} > \alpha) + p_i^s \big], \\
\tau_i^e &= \frac{1}{2} \big[ maxIdx(V_i^{mask} > \alpha) + p_i^e \big],
\end{aligned} \tag{8}
$$

where $\alpha$ is a threshold value. Then the video segment $V'$ between $(\tau_i^s, \tau_i^e)$ is the beneficial clip for current query.

**Turn Selection.** Based on $(\tau_i^s, \tau_i^e)$, the temporal relation between different QA turns can be explored. Since the attention of video dialog in different turns will shift, we consider the QA turns to be more relevant when they focus on close time regions. Therefore, we calculate the Intersection of Union (IoU) of timestamps between the current question and every history QA turns, and select the $k$ turns corresponding to the $k$ largest IoU:

$$\mathcal{H} = \text{top-k}\big[IoU[(\tau_{1:i-1}^s, \tau_{1:i-1}^e), (\tau_i^s, \tau_i^e)]\big], \tag{9}$$

where $|\mathcal{H}| = k$. When there are not enough QA turns or several QA turns have the same predicted timestamp, we preferentially choose the nearest QA pairs as the supplementary.

### 3.4 ANSWER GENERATION

Using the predicted $\mathcal{V}$ and $\mathcal{H}$, we can filter the irrelevant part of the whole video and the useless history turns. Specifically, we construct the video attention masks according to $(\tau_i^s, \tau_i^e)$ so that the attention weight of irrelevant video clips always equals to zero. At the same time, we only embed the relevant turns based on $\mathcal{H}$. After the same encoders as Sec. 3.2, the single modal features $\mathbf{T}$ and $\mathbf{V}$ can be expressed as $\mathbf{T_{use}}$ and $\mathbf{V_{use}}$. Then we utilize the same cross-modal encoder as Sec. 3.3 to obtain the fused feature $\mathbf{M_{use}} = CrossEncoder(\mathbf{T_{use}} \oplus \mathbf{V_{use}})$, where $\mathbf{M_{use}} \in \mathbb{R}^{(n'+m') \times d}$, $n' \leq n$ and $m' \leq m$.

Finally we adopt the decoder structure of Univl, which is a uni-directional attention model that generates the tokens one by one, to have the capability of learning from and benefiting the generation tasks. The decoded feature $\mathbf{D} \in \mathbb{R}^{l \times d_t}$ can be expressed as:

$$\mathbf{D} = Decoder(\mathbf{M_{use}}), \tag{10}$$

where $l$ is the decoder length, from which a sequence of words is generated as the system response and $d_t$ is the size of the token vocabulary. We employ the cross-entropy loss on the generated answers for model training:

$$L_{\text{gengerate}} = L_{\text{ce}}(\mathbf{D}, \mathbf{D_{gt}}), \tag{11}$$

where $\mathbf{D_{gt}}$ is the one-hot feature obtained from the ground truth response $A_K$. During evaluation, we use beam search to enhance the ability of generation, similar to other video dialog models.

### 3.5 CONTRASTIVE SELECTION

The utilization of cross-modal information can be enhanced by locating specific video clips according to each turn, and then spotting useful turns. However, not all QA turns can be accurately grounded. To solve this problem, we design a method inspired by contrastive learning (Liu et al., 2021c) to enhance the grounding ability between QA turns and video clips. We try to make the video dialog model more discriminative by pulling close positive samples $v^+$ and pushing away noisy negative samples $v^-$.

As shown in the right part of the **Contrastive Selection** in Fig. 2, for each video sample $v$, we nominate video clips between the range of $(\tau_i^s, \tau_i^e)$ as groundtruth sample $v_{gt}$ and video clips slightly larger than this range as poitive sample $v^+$. Correspondingly, video clips of other range are chosen as negative samples $v^-$. Similar to Sec. 3.4, we also construct video attention masks to obtain the required video clips. The features of the three samples can be expressed as $\mathbf{V_{use}}$, $\mathbf{V}^+$ and $\mathbf{V}^-$. Then, a MSE loss function is utilized to make the distance between the positive samples closer in the embedding space:

$$L^+ = MSE\big[\mathbf{M_{use}}, CrossEncoder(\mathbf{T_{use}} \oplus \mathbf{V}^+)\big].$$

We also utilize MSE loss function to make the distance between the positive samples and negative samples farther in the embedding space:

$$L^- = 1 - MSE\big[\mathbf{M_{use}}, CrossEncoder(\mathbf{T_{use}} \oplus \mathbf{V}^-)\big].$$

Then we can get the contrastive loss:

$$L_{\text{contrastive}} = L^+ + \beta L^-, \tag{12}$$

where $\beta$ is a hyperparameter to control the ratio. Finally, we utilize another hyperparameter $\delta$ and obtain the final loss of answer generation:

$$L_{\text{final}} = L_{\text{generate}} + \delta L_{\text{contrastive}}. \tag{13}$$

Table 1: Performance comparison (%) of DTGVD with SOTA methods on AVSD@DSTC-7 dataset. The best performance is marked in bold, and the second-best is underlined.

| Methods | CIDEr | BLEU-1 | BLEU-2 | BLEU-3 | BLEU-4 | METEOR | ROUGE-L | Avg |
|---|---|---|---|---|---|---|---|---|
| FA+HRED | 0.843 | 0.648 | 0.505 | 0.399 | 0.323 | 0.231 | 0.510 | 0.494 |
| MTN | 0.985 | 0.688 | 0.550 | 0.444 | 0.363 | 0.260 | 0.541 | 0.547 |
| Student-Teacher | 1.005 | 0.686 | 0.557 | 0.458 | 0.382 | 0.254 | 0.537 | 0.554 |
| VGD-GPT2 | 1.052 | 0.694 | 0.570 | 0.476 | 0.402 | 0.254 | 0.544 | 0.570 |
| BiST | 1.050 | 0.711 | 0.578 | 0.475 | 0.394 | 0.261 | 0.550 | 0.574 |
| SCGA | 1.059 | 0.702 | 0.588 | 0.481 | 0.398 | 0.256 | 0.541 | 0.575 |
| JST | 1.079 | - | - | - | 0.406 | 0.262 | 0.554 | - |
| COST | 1.085 | 0.723 | 0.589 | 0.483 | 0.400 | 0.266 | 0.561 | 0.587 |
| DTGVD (ours) | **1.152** | **0.732** | **0.604** | **0.508** | **0.423** | **0.271** | **0.571** | **0.606** |

## 4 EXPERIMENT

### 4.1 DATASETS

To evaluate the performance of our proposed DTGVD model, we conduct experiments on the challenging video grounded dialog dataset: Audio-Visual Scene-Aware Dialog (AVSD). It contains dialogs based on the Charades dataset (Sigurdsson et al., 2016). Each annotated dialog consists of up to 10 dialog turns. Each turn contains the question-answer pairs about objects, actions, events, and so on, and the corresponding reasoning timestamps in the video. AVSD dataset also contains three different testing splits, i.e. AVSD@DSTC-7 (Yoshino et al., 2019), AVSD@DSTC-8 (Kim et al., 2019) and AVSD@DSTC-10 (Hori et al., 2022). The training and validation sets are identical across all three splits, with AVSD@DSTC-10 additionally providing timestamp labels for each dialog turn. However, the test set for AVSD@DSTC-10 remains unpublished. In line with Le & Hoi (2020); Pham et al. (2022), we compare our method against other state-of-the-art approaches using the AVSD@DSTC-7 and AVSD@DSTC-8 test splits.

### 4.2 EVALUATION METRICS

Following existing video dialog works, we evaluate the performance on four main metrics: BLEU, METEOR, ROUGE-L and CIDEr, which are widely used such as by Le & Hoi (2020); Pham et al. (2022) to evaluate the performance of the proposed methods. We also calculate the average of all metrics to assess the overall performance. Besides, we adopt "R@$n$, IoU $= \mu$" to evaluate the temporal duration of each question-answer turn, following Gao et al. (2017). The "R@$n$, IoU $= \mu$" represents the percentage of language queries having at least one result whose IoU between the top-$n$ predictions with the ground-truth is larger than $\mu$. In our experiments, we reported the results of $n = 1$ and $\mu \in \{0.3, 0.5, 0.7\}$.

**Human Evaluation.** As Hori et al. (2022), we employed a 5-point Likert scale to gather human ratings for each system response. Human raters evaluated system responses under given dialogue context and video conditions, where a score of 5 indicated excellent, 4 denoted good, 3 represented acceptable, 2 signified poor, and 1 indicated very poor quality. Human raters were instructed to primarily focus on two aspects: the accuracy of answers considering the context and video, and the fluency of the responses.

### 4.3 IMPLEMENTATION DETAILS

For the structure of pretrained model, we follow the implementation of UniVL (Luo et al., 2020), which contains 12 Transformer layers for text encoder, 6 Transformer layers for visual encoder, 2 Transformer layers for cross-modal encoder, and 3 Transformer layers for decoder part. A fine-turned UniVL is used as baseline for comparison. All datasets are trained for 8 epochs till converge. We use Adam optimizer with a initial learning rate of 3e-5, and a batch size of 128 samples distributed on 2 Nvidia Tesla V100 GPUs with 32GB memory. For video features, we adopt the S3D model (Xie et al., 2018b) which outputs a 1024-dimensional vector. After obtaining embeddings of video and text, we concatenate three embeddings in the following sequence: video, current question

and dialog history, and limit the length of each embedding to 100, 20 and 60, respectively. For hyperparameters mentioned in Sec. 3, we set threshold $\alpha = 0.5$, maximum history turns $k = 3$, loss control ratio $\lambda = 0.2$, $\beta = 0.5$ and $\delta = 0.2$ in our experiment. The whole system is implemented with PyTorch framework. More details can be found in our code.

Table 2: Performance comparison (%) of DTGVD with SOTA methods on AVSD@DSTC-8 dataset.

| Methods | CIDEr | BLEU-1 | BLEU-2 | BLEU-3 | BLEU-4 | METEOR | ROUGE-L | Avg |
|---|---|---|---|---|---|---|---|---|
| MTN | 0.912 | 0.643 | 0.523 | 0.427 | 0.356 | 0.245 | 0.525 | 0.519 |
| VGD-GPT2 | 1.022 | 0.677 | 0.556 | 0.462 | 0.387 | 0.249 | 0.544 | 0.557 |
| SCGA | 1.024 | 0.675 | 0.559 | 0.459 | 0.377 | 0.269 | 0.555 | 0.560 |
| JST | 0.997 | - | - | - | 0.394 | 0.250 | 0.545 | - |
| COST | 1.051 | 0.695 | 0.559 | 0.465 | 0.382 | **0.278** | **0.574** | 0.572 |
| DTGVD (ours) | **1.076** | **0.705** | **0.582** | **0.482** | **0.402** | 0.264 | 0.567 | **0.583** |

## 4.4 PERFORMANCE COMPARISON AGAINST SOTA

Some SOTA methods utilize extra information of video, such as caption, subtitle, and so on. However, these additional data sources are not always accessible in real application. To make a fair comparison, we only take video content and dialog history as input.

We mainly make the comparison with the following state-of-the-art methods: JST (Shah et al., 2022), VGD-GPT2 (Le & Hoi, 2020), SCGA (Kim et al., 2021), MTN (Le et al., 2019b), FA+HRED (Nguyen et al., 2019), Student-Teacher (Hori et al., 2019b), BiST (Le et al., 2020), and COST (Pham et al., 2022). Among them, Student-Teacher (Hori et al., 2019b) and JST (Shah et al., 2022) utilize teacher model to obtain additional information from summary. SCGA (Kim et al., 2021) and COST (Pham et al., 2022) employ extracted object features to interact with text. FA+HRED (Nguyen et al., 2019), MTN (Le et al., 2019b) and BiST (Le et al., 2020) use multiple attention for cross-modal fusion. VGD-GPT2 (Le & Hoi, 2020) inherits the embedding and text generation capabilities of pre-trained model. The performances of other SOTA methods are reported according to their respective papers or by running their released codes.

As shown in Table 1, DTGVD achieves the best performance across all metrics on AVSD@DSTC-7. Compared with the current SOTA method COST, DTGVD achieves 5.8% improvement (0.423 vs 0.400) in BLEU-4, and 5.5% improvement (1.145 vs 1.085) in CIDEr. On AVSD@DSTC-8, results are reported in Table 2. DTGVD still shows performance improvement on 6 out of 8 metrics compared with other SOTA (1.076 vs 1.051 in CIDEr). Among these metrics, BLEU focuses on precision, ROUGE-L emphasizes recall, METEOR considers both, and CIDEr pays more attention to key information. Due to more accurate utilization of useful information in both video and history, the answers generated by DTGVD are more capable of filtering out irrelevant information and focusing on key information in relevant history. Therefore, it leads to a significant improvement in BLEU and CIDEr. For other existing SOTA methods, using the entire video and all history turns (or several recent history turns) often leads to the inclusion of interference information in the generated answers, resulting in significant deficiencies in BLEU and CIDEr.

The removal of irrelevant information by DTGVD inevitably results in answers that focus more on key information, but lack some less useful words that can improve recall. This results in some "unreal" deficiencies in METEOR and ROUGE-L for DTGVD in AVSD@DSTC8. Therefore, we added Avg to represent the average of all metrics to reduce the impact of shortcoming of a single evaluation method. Avg results indicate that DTGVD has significant advantages on both datasets.

Additionally, we conducted human evaluation comparing our model to the current SOTA model, COST (Pham et al., 2022), to further validate the evaluation results. In terms of fluency, DTGVD scored 4.221 while COST scored 4.109. In terms of accuracy, DTGVD scored 3.678 while COST scored 3.237. The greater enhancement in accuracy can be attributed to DTGVD's refined emphasis on related segments within both text and video.

Table 3: Abaltion studies of different components in DTGVD model (UniVL) on AVSD@DSTC-7.

| Components | | | CIDEr | BLEU-4 | METEOR | ROUGE-L |
| Turn Selection | Video Mask | Contrastive | | | | |
|---|---|---|---|---|---|---|
| | | | 1.092 | 0.407 | 0.260 | 0.557 |
| ✓ | | | 1.113 | 0.406 | 0.264 | 0.558 |
| ✓ | ✓ | | 1.137 | 0.416 | 0.268 | 0.566 |
| ✓ | ✓ | ✓ | **1.145** | **0.423** | **0.271** | **0.571** |

## 4.5 ABLATION STUDIES

We design multiple ablation experiments to explore the impact of each component of the proposed method, including the pre-trained models, contrastive selection, video mask and history QA turns selection. The experiments show that each component has a positive impact on the final results, as shown in Table 3.

**The effect of temporal grounding.** Our proposed temporal grounding mechanism includes two aspects: the selection of dialog history turns and the highlighted video features. For the former, if we choose the related history QA pairs according to the timestamps, the performance of baseline model will increase from 1.092 to 1.113 (1.9%) in CIDEr. For the later, if we block irrelevant clips, the performance will increase from 1.113 to 1.137 (2.2%) in CIDEr, compared with inputting visual feature with whole video sequence. Experimental results show that both the selection of dialog history turns and highlighted video features are beneficial to the final performance.

**The effect of contrastive selection.** According to Table 3, contrastive selection brings a 0.7% boost in CIDEr (from 1.137 to 1.145). Note that this method is employed to highlight related video clips more accurately. Thus, the effectiveness of contrastive selection also demonstrates that DTGVD still has the potential for improvement, if the grounding model is more reliable.

## 4.6 TEMPORAL GROUNDING PERFORMANCE

Since the test set of AVSD@DSTC-10 includes timestamp labels but is not public, we cannot compare with existing results. Instead, we evaluate temporal accuracy on the AVSD@DSTC-10 validation set, where our DTGVD achieves competitive performance with 0.728 in R1@0.3, 0.652 in R1@0.5, and 0.544 in R1@0.7 for the video grounding task.

## 4.7 PERFORMANCE ON VARIOUS PRETRAINED MODEL

The experiments in the previous sections are all conducted using DTGVD with UniVL as the baseline. However, the methods used in DTGVD can also be transferred to various pretrained models, and yielding performance improvements. Table 4 shows the percentage increase in CIDEr after applying the proposed methods to GPT-2 (Radford et al., 2019), LLaMA (Touvron et al., 2023) and UniVL (Luo et al., 2020). We mimic the video processing methods from VGD-GPT2 (Le & Hoi, 2020) and Video-LLaMA (Zhang et al., 2023) for GPT-2 and LLaMA, respectively, serving as comparative baselines. Upon this foundation, we apply the principal methods proposed herein to them, i.e., Turn Selection, Video Mask, and Contrastive Selection. Then we calculate the percentage improvement in CIDEr scores relative to the baseline upon application of these methods.

Table 4: Improvement of CIDEr of different pretrained models with the proposed method.

| Pretrained Model | Pretraining Modalit | Params (B) | Improvement in CIDEr(%) |
|---|---|---|---|
| GPT-2 | Text | 1.5 | 1.7 |
| LLaMA | Text | 7 | 4.2 |
| UniVL | Text-Video | 0.13 | 4.9 |

It is observed that all three pretrained models show performance gains with the proposed approach, with UniVL demonstrating the largest improvement, likely due to its multimodal text-video pretraining, enhancing text-video interactions. GPT-2 and LLaMA, originally pretrained on text only

and adapted for video via an additional encoder, may have a less comprehensive understanding of video content. LLaMA, with its larger parameter set, exhibits greater improvement. Thus, further improvements in the DTGVD framework could be achieved by enhancing text-video interaction capabilities or using more powerful pretrained models.

## 4.8 IN-DEPTH ANALYSIS

**Q1: What if the predicted temporal region is inaccurate?** It is evident that not all question-answer pairs have an exact corresponding video clip. Particularly, for complicated questions that require multiple steps of reasoning, the predicted temporal region may not be entirely precise. In such cases, the grounding model often predicts more frames than necessary. To address this issue, we consider extended regions as positive samples to minimize the adverse effects of inaccurate grounding. As a result, even if the predicted region is longer than the actual region, their encoded features will remain relatively consistent.

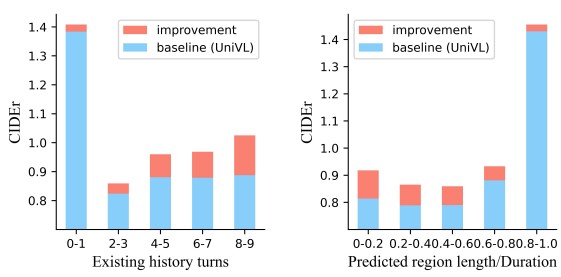

(a) Effect of history turns    (b) Effect of region length

Figure 3: The CIDEr performance of DTGVD and baseline (UniVL) with regard to different number of existing history turns and different length of predicted video region.

**Q2: Is the history turn selection really useful?** Fig. 3 (a) shows the different CIDEr performance of DTGVD and baseline under various number of history turns. For example, if history turns are 6, it means the current question is the 7-th turn. In the case that we select three most related turns, the more history turns exist before current question, the performance difference between the two models would theoretically be larger. The results in the figure confirm our estimate. Besides, the closest three turns are chosen for baseline model. So when the number of history turns is less than three, there should be little difference in performance between the two models. Indeed, we can observe that there is a huge change when there are less or more than three history turns.

**Q3: Is the video mask really useful?** Just like **Q2**, Fig. 3 (b) shows the different CIDEr performance of DTGVD and baseline under various length of predicted region. If the proportion of the predicted region to the video duration is smaller, it means that more irrelevant regions are blocked. We can notice that as the ratio gets smaller, the CIDEr improvement ratio gets higher between the two models. This further illustrates the effectiveness of the video mask, and all the experiments above prove that **Grounding is All You Need in Video Dialog**.

## 5 CONCLUSION

To enhance the filtering capability of both visual and textual information simultaneously for video dialog, this paper proposes a Dual Temporal Grounding-enhanced Video Dialog model (DTGVD), which utilizes the pre-trained visual-language model and excludes irrelevant video clips and dialogue history turns based on the predicted temporal area of each question-answer pairs, thus making the answers in video dialogue more accurate. We also choose accurately grounded turn-clip pairs as positive samples and gather other turn-clip pairs as negative samples in order to better illustrate the temporal relationship between the two modalities. The entire model is then trained using answer generation loss and contrastive learning loss. Experiments on two well-known benchmark datasets demonstrate the effectiveness of our proposed method. And experiments on various pretrained models verified the adaptability of the method.

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

## A    APPENDIX

## B    DATASET

We conduct experiments on Audio-Visual Scene-Aware Dialog (AVSD) to evaluate the results. This dataset contains shared training/validation set, and two different test sets, namely AVSD@DSTC-7 and AVSD@DSTC-7. Details of the dataset are shown in Table 5.

## C    IN-DEPTH ANALYSIS

### C.1    TEMPORAL GROUNDING PERFORMANCE

"R@$n$, IoU = $\mu$" is a common metric for evaluating grounding performance. But IoU cannot fully demonstrate the validity of results in the task setting. For example, predicting full-length video as a positive region may also result in a relatively large IoU, but it cannot block irrelevant regions. Even if the indicators on the validation set are higher than those of other SOTA grounding models, it cannot fully demonstrate that our grounding results on the test set is good enough.

Thus, in Figure 4, we compare the groundtruth of temporal regions in the training dataset with the predicted ones in the test set of AVSD@DSTC-7 and AVSD@DSTC-8. Specifically, the horizontal axis represents the ratio of timestamp to video duration, and the vertical axis represents the percentage of frames in this ratio. For example, if the whole video length is 10s, the useful region is between 2s and 5s and the number of all frames is 10000, then the vertical coordinate value corresponding to the horizontal coordinates of 0.2 to 0.5 are added by 0.01%. As the test set does not have timestamp labels, if the predicted results are similar to the distribution of the groundtruth of training set, it signals that our grounding results are effective. As shown in Figure 4, the distributions of the two are indeed very similar.

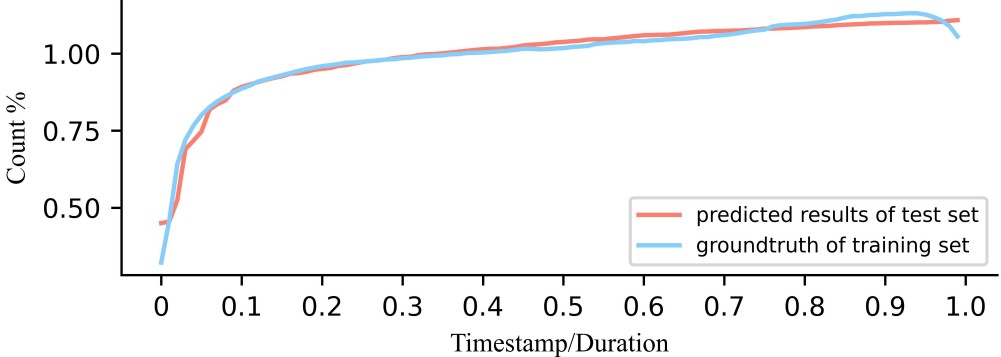

Figure 4: The distribution of temporal regions within the training set's ground truth and the predicted regions from the test set.

### C.2    MODALITY OF CONTRASTIVE SELECTION

Upon realizing that contrastive learning can have a positive impact, it is easy to consider creating positive and negative text samples. For instance, unselected turns could be used as negative samples.

|                | Training | Validation | DSTC7 Test | DSTC8 Test |
|----------------|----------|------------|------------|------------|
| # Video        | 7659     | 1787       | 1710       | 1710       |
| # Dialog turns | 153180   | 35740      | 13490      | 18810      |

Table 5: Statistics of the AVSD dataset.

| Contrast pairs | | BLEU-4 | METEOR | ROUGE-L | CIDEr |
| $\mathbf{V}^{+/-}$ | $\mathbf{T}^{+/-}$ | | | | |
| --- | --- | --- | --- | --- | --- |
| | | 0.416 | 0.268 | 0.566 | 1.137 |
| ✓ | | 0.423 | 0.271 | 0.571 | 1.145 |
| | ✓ | 0.412 | 0.264 | 0.561 | 1.121 |
| ✓ | ✓ | 0.417 | 0.266 | 0.562 | 1.133 |

Table 6: Performance of DTGVD with different contrast pairs.

However, this may not be beneficial for two reasons. Firstly, the aim in using contrastive learning is to improve temporal grounding accuracy. Nevertheless, incorrect positioning will only affect the selection of video clips, and the relationship between turns will remain unchanged. To put it simply, turns with high temporal overlap will still have a large IoU, even if the grounding is imprecise. Secondly, creating negative text examples may have an adverse effect on the results. In this task, only the relevant video clip, current question, and answer are highly correlated, not the history turns. In other words, relevant history turns improve the answer, but irrelevant turns should not be expected to make the answer worse. As shown in Table 6, we compare the performance of DTGVD with different contrast pairs, where $\mathbf{T}^{+/-}$ means adding one more history turns as positive samples, i.e. $k = 4$, and utilizing the remaining irrelevant history turns as negative examples. The results indicate that $\mathbf{V}^{+/-}$ improves the performance while $\mathbf{T}^{+/-}$ has a negative impact.

## D  QUALITATIVE ANALYSIS

We further perform qualitative analysis on the method to enable a better understanding of its strength. Figure 5 visualizes the working process of DTGVD with a sample from AVSD@DSTC-7 dataset. Through temporal grounding model, the predicted timestamps of each turns are first obtained. The region corresponding to current question is 13.31s to 21.02s. After calculating IoU between timestamps of each turns and that of current question, the DTGVD model selects the three turns with highest IoU, i.e. Q2&A2, Q3&A3 and Q4&A4. Conversely, the baseline model UniVL selects the most recent three turns, i.e. Q4&A4, Q5&A5, and Q6&A6, and the whole video as input. The results indicate that the baseline answer is affected by irrelevant motion before the man looking at his phone and does not focus on the correct temporal position. Additionally, Q5&A5 and Q6&A6 are not related to Q7 and only add noise to the answer generation. In contrast, DTGVD excludes video clips before the man looking at something to avoid ambiguity and utilizes Q3&A3 to confirm the information about the man walking to another room. The effectiveness of the selection of history turns and video clips is demonstrated by the quantitative results.

In Figure 6 and Figure 7, we provide visualizations for different scenarios. Figure 6 mainly shows that when there are few history turns, the text input of DTGVD and the baseline are exactly the same. However, DTGVD blocks out irrelevant video clips, which has a positive impact on the answer. Figure 7 mainly shows that when the current question is difficult to be accurately grounded, DTGVD can still find relevant history turns, thereby obtaining more useful information when answering. The visualizations in both scenarios further demonstrate the effectiveness of DTGVD.

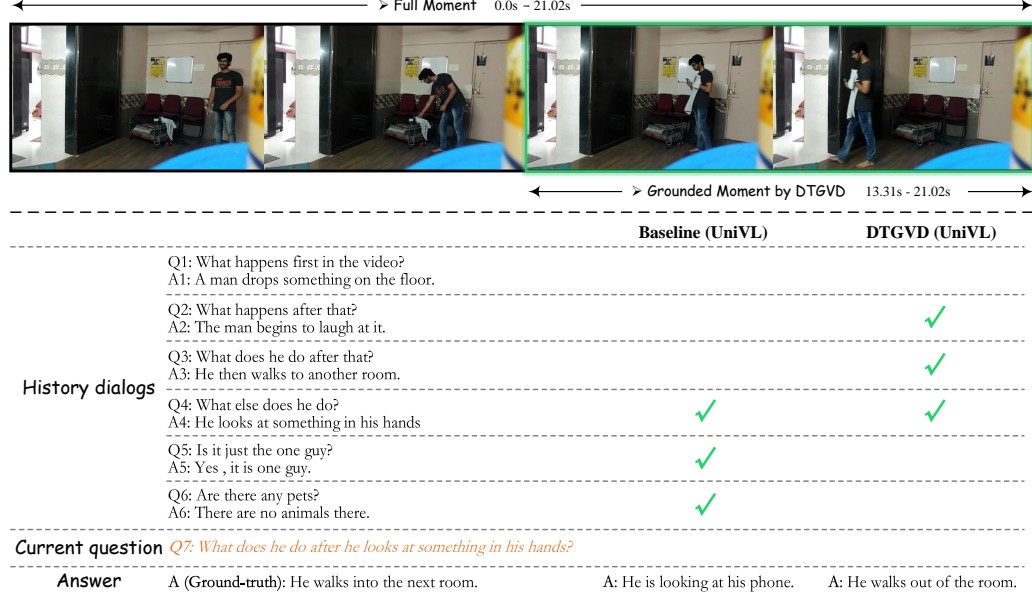

Figure 5: Qualitative results of DTGVD on AVSD@DSTC-7 dataset. The QA turns selected with green check marks are actually used by the model, which means DTGVD utilizes Q2&A2, Q3&A3 and Q4&A4, and UniVL utilizes Q4&A4, Q5&A5 and Q6&A6. The video clips framed in green are actually used by DTGVD, while the whole video is used by UniVL.

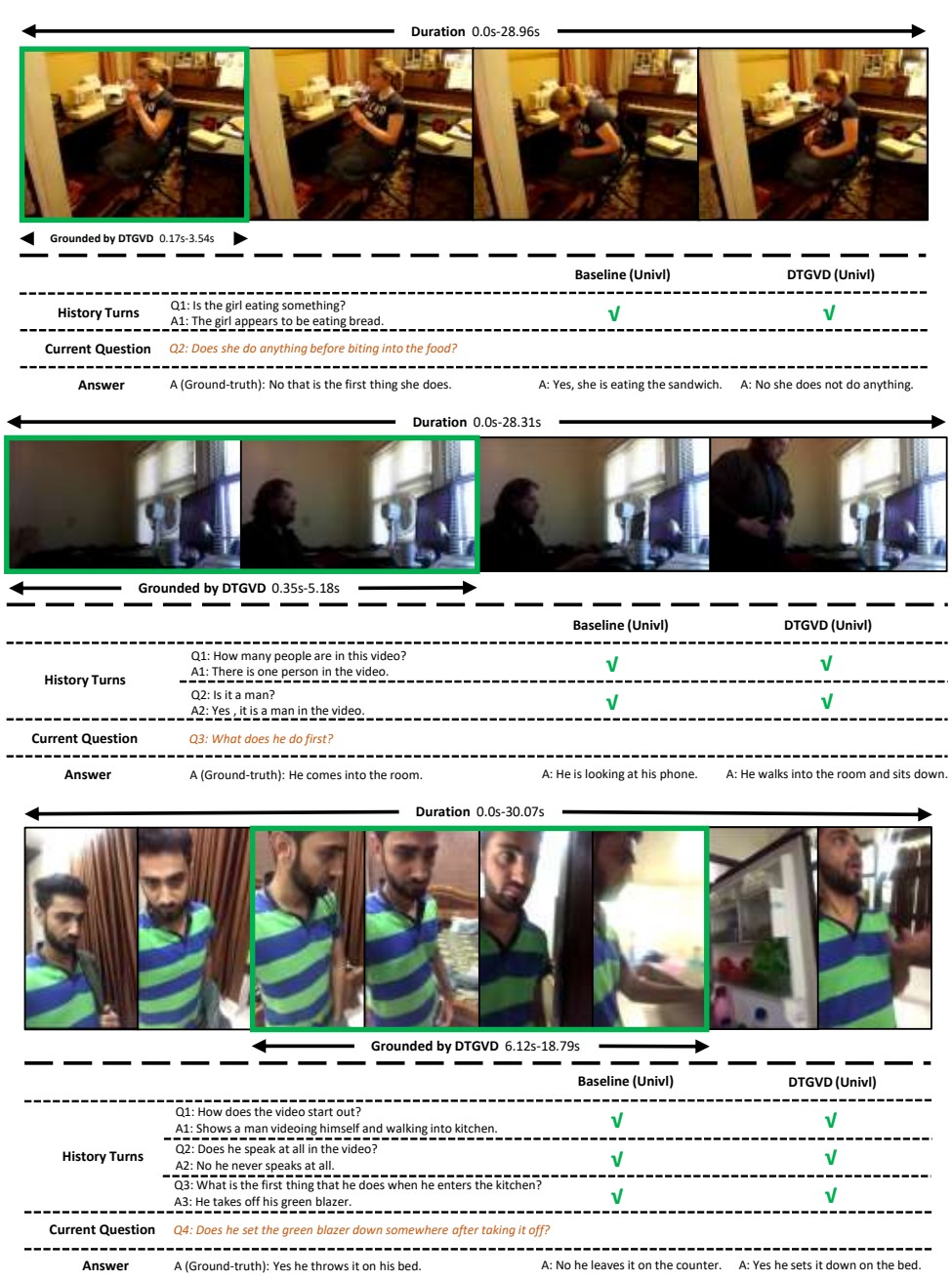

Figure 6: Visualization with few history turns. The text input is identical between the two models, but DTGVD improves answer accuracy by selecting relevant video clips.

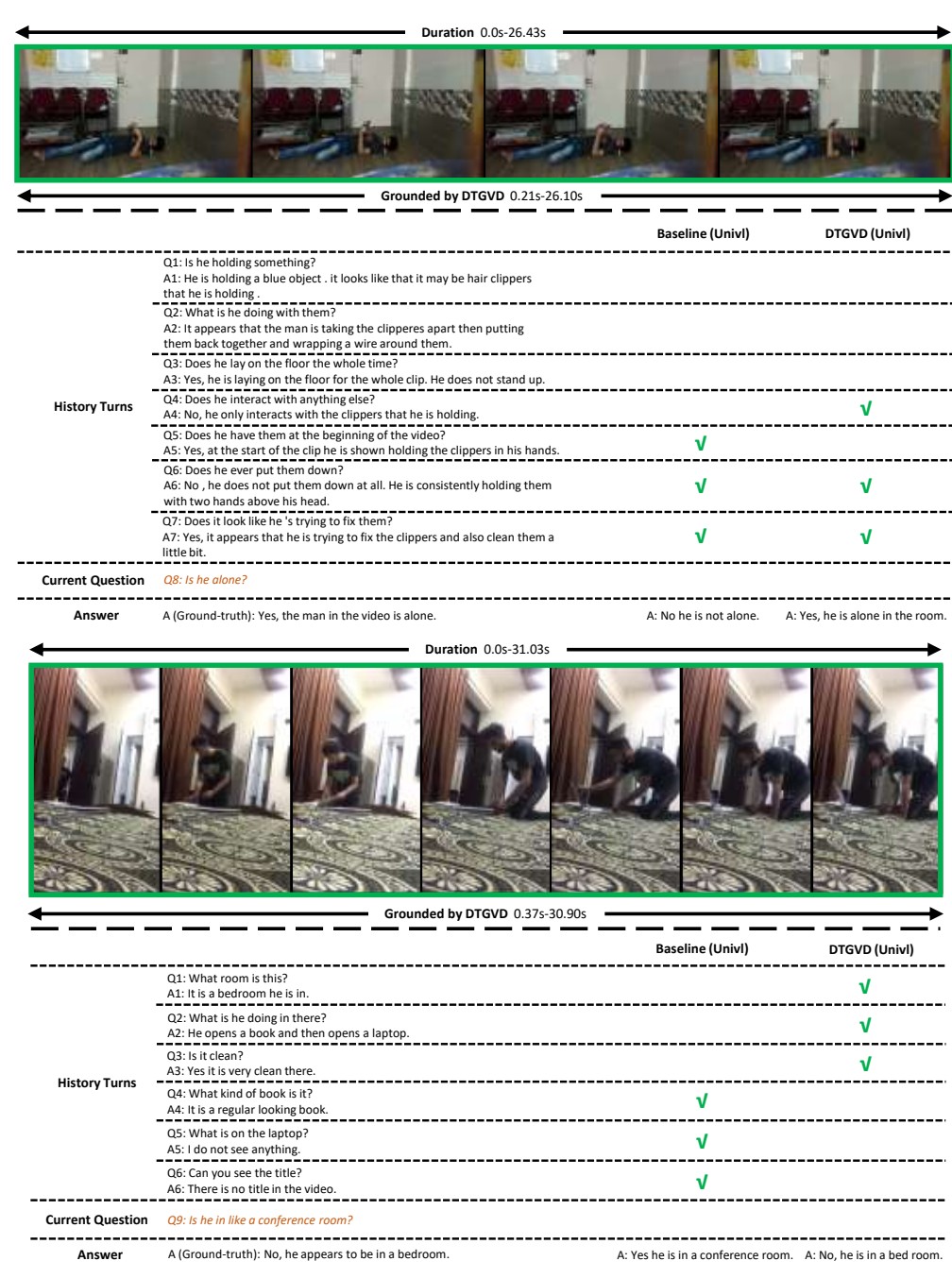

Figure 7: Visualization when the timestamp corresponding to the current question cannot be found. Both models input the full-length video, but DTGVD improves answer accuracy by selecting relevant history turns.

