# OpenReview forum: "Grounding is All You Need? Dual Temporal Grounding for Video Dialog"
_ICLR.cc/2025/Conference — ICLR 2025 Conference Withdrawn Submission_

### Official Review · Reviewer_nnk5 · 2024-10-30

**Soundness:** 2
**Presentation:** 2
**Contribution:** 2
**Rating:** 3
**Confidence:** 5

**Summary:**

This paper presents the Dual Temporal Grounding-enhanced Video Dialog model (DTGVD), which integrates video content and conversation history to enhance response generation. By aligning turn-specific temporal regions in both video and dialog, DTGVD improves the understanding of conversational dynamics. Tested on AVSD@DSTC7 and AVSD@DSTC8, DTGVD effectively captures temporal dependencies and generates accurate responses.

**Strengths:**

1.The idea of the paper is easy to understand.
2.The presentation of the method section is relatively detailed.

**Weaknesses:**

1.The idea of this article is similar to [1] in that both are trying to mine meaningful historical information. Can you clarify the difference between the two? And the advantages and disadvantages?
2.Too many recent baseline methods in this field are not cited or compared, e.g., ISR[1], PDC[2], RLM[3], T5RLM[4], JVIT[5], and DialogMCF[6]. Also, why are the baseline results in your paper inconsistent with the results in other papers? For example, SCGA obtains 0.745 in BLEU-1 in [1] vs 0.702 in your paper.
3.More diverse evaluation datasets are desired.The training and validation sets for DSTC7 and DSTC8 are identical and can be considered as one dataset.
4.Where are the results of the temporal duration (R@n, IoU) and human evaluation mentioned in the paper? I can't find them in the main text or in the appendix.

[1] Zhang et al. Uncovering Hidden Connections: Iterative Search and Reasoning for Video-grounded Dialog. 2023.
[2] Le et al. Learning Reasoning Paths over Semantic Graphs for Video-grounded Dialogues. 2021.
[3] Li et al. Bridging Text and Video: A Universal Multimodal Transformer for Audio-Visual Scene-Aware Dialog. 2021.
[4] Yoon et al. Information-Theoretic Text Hallucination Reduction for Video-grounded Dialogue. 2022.
[5] Zhang et al. Video-Grounded Dialogues with Joint Video and Image Training. 2022.
[6] Chen et al. DialogMCF: Multimodal Context Flow for Audio Visual Scene-Aware Dialog. 2023.

**Questions:**

See above.

---

### Official Review · Reviewer_BtMD · 2024-11-03

**Soundness:** 2
**Presentation:** 1
**Contribution:** 2
**Rating:** 5
**Confidence:** 3

**Summary:**

This paper introduces the Dual Temporal Groundingenhanced Video Dialog model (DTGVD) for video dialog response generation. This paper filters out the irrelevant video clips and dialog history through a temporal grounding module. Further, a contrastive learning method is proposed to utilize the relative video and dialog history for better video dialog generation. The results prove the effectiveness of this method.

**Strengths:**

1. This paper proposes a Dual Temporal Groundingenhanced Video Dialog model to achieve SOTA performance on the video dialog generation task.
2. This paper proposes comprehensive experiments to prove the effectiveness of this method.

**Weaknesses:**

1. The writing of this paper needs further improvement. For example, the abstract contains several sentences that lack clarity, and there is a lack of coherence between the different parts of the abstract.
2. The selection of dialog history relies on the timesteps of each QA pair. In order to be applicable in real-world scenarios, the timestamps of this QA should not be considered accessible information for model inference.
3. The contrastive learning method is simply utilized to optimize the textual and vision feature encoder. Could this method be further utilized to refine the selection results of vision and dialog history?
4. This method should be compared with some Multi-modal Large Language models method.

**Questions:**

Please refer to the weaknesses

---

### Official Review · Reviewer_npFV · 2024-11-04

**Soundness:** 3
**Presentation:** 2
**Contribution:** 3
**Rating:** 3
**Confidence:** 4

**Summary:**

It presents the DTGVD model for video dialog response generation. It combines the strengths of video content understanding and conversation history analysis through dual temporal grounding. The model predicts relevant temporal regions in videos and dialogs, filters content, and uses contrastive learning to align video and dialog dynamics. Experiments on AVSD@DSTC datasets show improved performance.

**Strengths:**

1. It introduces an innovative approach with the Dual Temporal Grounding-enhanced Video Dialog (DTGVD) model, which leverages dual temporal dynamics inherent in both video sequences and dialog histories.
2. This model employs a temporal grounding module to explicitly model the attention shift of each dialog turn over the video, generating temporal masks to filter out irrelevant video frames and dialog history.

**Weaknesses:**

1. The motivation is not convincing enough. Why the temporal information in the dialog history is needed? Existing works can also model the video temporal structure and answer the question.

2. the baseline models are too old and they are not the latest SOTA. There are some new works for example:
[1] HEAR: Hearing Enhanced Audio Response for Video-grounded Dialogue
[2] M2K-VDG: Model-Adaptive Multimodal Knowledge Anchor Enhanced Video-grounded Dialogue Generation

The grounded QA methods should also be compared. Since it solves the same questions essentially, like works
[3] Can I Trust Your Answer? Visually Grounded Video Question Answering

**Questions:**

1. Since the information provided by the dialog history is from the video, why not directly find the important frames in videos and learn QA from the video and question? The dialog has noises, so why the dialog history is needed to solve this QA?
2. In Figure 2, in the "contrastive-enhanced answer generation" module, why does the "exclude" branch link to the generation decoder?
3. Since you select the top k relevant question-answer pair to help answer the final question. Why do you select top-k from the IoU other than directly searching the relevant pair from the question semantics?

---

### Official Review · Reviewer_qtHC · 2024-11-05

**Soundness:** 3
**Presentation:** 2
**Contribution:** 2
**Rating:** 5
**Confidence:** 3

**Summary:**

Previous approaches for video dialog have rarely grounded relevant video frames, and more specifically, relevant dialog turns for generating a response to the current question. Most of them use merely previous dialog turns or full dialog, and also full video, which might lead to suboptimal performance. This paper proposes predicting the start and end timestamps of each dialog turn and the current question, and only using the video frames and dialog turns in the same range as the current question for generating the response. Furthermore, to enhance the relevant video clip selection, the proposed method, Dual Temporal Grounding-enhanced Video Dialog (DTGVD), is trained with a newly designed contrastive loss as well as answer generation loss. The empirical results on AVSD@DSTC-7 and AVSD@DSTC-8 demonstrate superiority of the proposed method.

**Strengths:**

- The idea of temporally grounding both relevant dialog turns and relevant video frames makes sense.
- The proposed method shows superior performance on AVSD@DSTC-7/DSTC-8, outperforming pervious approaches.
- The ablation studies and in-depth analysis validate the idea and model design choices.

**Weaknesses:**

The paper is overall good, but I have the following concerns:

- Recently, many video multimodal large language models (VideoLLMs) have been proposed, including Video-LLaVA [1], SeViLA [2] and LLaMA-VID [3], Video-ChatGPT [4] and Video-LLaMA. Please compare with them, if possible.
- Are there any other video dialog datasets that are sourced from different video distribution? Evaluating on other datasets such as VSTAR [5] would make the paper more powerful.
- In section 4.4 (L423-425), the authors mentioned that the removal of irrelevant information by the proposed method might lead to performance drop in METEOR and ROUGE-L. However, according to Table 3, removing irrelevant information by adding turn selection and video mask components improves METEOR and ROUGE-L scores, which contradicts this statement. Please explain about this.

And although the authors provided code, there are missing details in the paper:
- Is the training conducted in two stages? First, with L_grounding loss in Equation 7, and then with L_final in Equation 13?
- When computing the grounding loss, did the authors compute the loss for questions from the dialog history in addition to the current question? And how to compute the start and end timestamps during inference?
- L253: How did the authors compute logits for the cross-entropy loss?
- L273-L274: just selecting the top K, without any thresholding?
- L297: Did the authors try temperature sampling instead of beam search?
- The authors mentioned video "clips" in Section 3.5. Are they different from the untrimmed video V in Section 3.1? I am not sure how the authors sampled video clips to construct positive and negative samples for contrastive learning.

[1] Lin et al., Video-LLaVA: Learning United Visual Representation by Alignment Before Projection, EMNLP 2024.
[2] Yu et al., Self-chained image-language model for video localization and question answering, NeurIPS 2024.
[3] Li et al., LLaMA-VID: An Image is Worth 2 Tokens in Large Language Models, ECCV 2024.
[4] Maaz et al., Video-ChatGPT: Towards Detailed Video Understanding via Large Vision and Language Models, ACL 2024.
[5]  Wang et al., VSTAR: A Video-grounded Dialogue Dataset for Situated Semantic Understanding with Scene and Topic Transitions, ACL 2023.

**Questions:**

Please address the above concerns. I am willing to raise my rating based on the rebuttal.

---

### Note · Authors · 2024-11-13

I have read and agree with the venue's withdrawal policy on behalf of myself and my co-authors.